# Tension-Compression Fatigue Induced Stress Concentrations in Woven Composite Laminate

**Eldho Mathew** [1], **Rajaram Attukur Nandagopal** [1] , **Sunil Chandrakant Joshi** [1,*] , **Pinter Armando** [2] **and Pasi Matteo** [3]

1   School Mechanical and Aerospace Engineering, Nanyang Technological University,
    Singapore 639798, Singapore; eldho003@e.ntu.edu.sg (E.M.); rajaram003@e.ntu.edu.sg (R.A.N.)
2   Fatigue Dept., Leonardo Helicopter Division, I-21017 Cascina Costa, VA, Italy;
    armando.pinter@leonardocompany.com
3   Airframe & Systems Dept., Leonardo Helicopter Division, I-21017 Cascina Costa, VA, Italy;
    matteo.pasi@leonardocompany.com
*   Correspondence: mscjoshi@ntu.edu.sg

**Abstract:** Tension-compression (T-C) fatigue response is one of the important design criteria for carbon-fibre-reinforced polymer (CFRP) material, as well as stress concentration. Hence, the objective of the current study is to investigate and quantify the stress concentration in CFRP dog-bone specimens due to T-C quasi-static and fatigue loadings (with anti-buckling fixtures). Dog-bone specimens with a $[(0/90),(45/-45)4]_s$ layup were fabricated using woven CFRP prepregs and their low-cycle fatigue behaviour was studied at two stress ratios ($-0.1$ & $-0.5$) and two frequencies (3 Hz & 5 Hz). During testing, strain gauges were mounted at the centre and edge regions of the dog-bone specimens to obtain accurate, real-time strain measurements. The corresponding stresses were calculated using Young's moduli. The stress concentration at the specimen edges, due to quasi-static tension, was significant compared to quasi-static compression loads. Furthermore, the stress concentration increased with the quasi-static loading within the elastic limit. Similarly, the stress concentration at the specimen edges, due to tensile fatigue loads, was more significant and consistent than due to compressive fatigue loads. Finally, the effects of the stress ratio and loading frequency on the stress concentration were noted to be negligible.

**Keywords:** tension-compression fatigue; carbon fibre reinforced polymer; stress concentration; dog-bone specimen

## 1. Introduction

Over the years, lightweight and high-strength composite materials have gradually replaced conventional isotropic materials in the aerospace, marine and automotive industries [1]. The superior design tailorability of composite materials, due to their inhomogeneity and potential variety in its constituent materials and orientation, is one of their several benefits [2]. However, such design versatility also demands extensive coupon and component level testing to ensure certification compliance [3]. Changes to the constituent materials or the orientation of the load-carrying fibres in a composite laminate must be tested comprehensively to characterize the static and dynamic behaviour of the laminate.

The fatigue response of a composite laminate is one of the essential design criteria, which is considered during any structural certification exercise. ASTM and ISO standard fatigue tests are performed to characterize the fatigue behaviour of the laminate at various environmental conditions [4]. At the coupon level, these tests are conducted on small specimen sizes resulting in the overestimation of the laminate fatigue strength due to size effects [5,6]. Thus, the design of a large structure based on coupon tests should take the size effect into account. In industries, large coupon testing is performed to partially overcome the size effects. In the case of axial loading, such large coupons are dog-bone shaped

which might introduce stress concentrations in the specimen. The primary objective of this study is to quantify the stress concentrations in large dog-bone specimens subjected to quasi-static and axial fatigue loadings.

The fatigue behaviour of composite materials was comprehensively investigated in the past, especially the tension-tension (T-T) fatigue behaviour [7,8]. In these studies, the S-N and ε-N plots of numerous composite laminates at different conditions were established [9,10]. Typically, the residual strength of the laminates is experimentally measured, while analytical, empirical, and numerical models are developed for fatigue life prediction [11]. Furthermore, stress concentrations in notched composite specimens were also reported due to the T-T fatigue loading [12]. Investigating the fatigue behaviour under T-T loading is relatively simple due to the uncomplicated nature of the loading mechanism. However, the tension-compression (T-C) and compression-compression (C-C) fatigue loadings on large samples are complicated due to the involvement of anti-buckling rods that prevent buckling during loading [13]. Due to this complexity, the T-C and C-C fatigue studies are usually performed on specimens with very small gauge lengths (~15 mm) [14]. Hence, even the basic fatigue data (such as S-N data) of large composite specimens due to T-C loading are rarely found in the open literature. Additionally, to the best of the authors' knowledge, the stress concentration data of large composite specimens due to T-C fatigue loading are completely lacking. Hence, in this study, the effect of stress concentration on large dog-bone specimens was investigated under T-C fatigue loading. Furthermore, the frequency and stress ratio of the loading were varied to identify any possible effects of these parameters on the stress concentration.

tension-compression fatigue is a common in-service loading experienced by aircraft composite structural members. It is relatively more detrimental to multi-directional composite laminates with off-axis plies than T-T fatigue loading [14–16]. The presence of transverse plies, in particular, deteriorates the laminate significantly due to T-C loading [17]. The local fatigue failure of the matrix during T-C loading induces fibre splitting and delamination failure in the laminate due to stress concentration, leading to its deterioration [15]. Furthermore, the alternate fatigue loading in tension and compression may increase the probability of crack openings and propagation compared to tension-only fatigue loading [18]. This is due to the opening of microcracks along the loading direction, induced by global or local buckling. Again, the possibility of the above-mentioned failure mechanism occurring at multiple locations increases in large composite specimens due to the size effects. In addition to the local stress concentrators near microcracks, the geometric nonlinearity of the specimen itself may act as stress concentrators, and thus aggravate the fatigue failure. Hence, the combination of size effects (large specimen), T-C fatigue loading and local stress concentration at microcrack tips is very detrimental to composite structures. Since the latter two are usually studied using small composite specimens, the stress concentration due to geometric non-linearity and size effects are often inadvertently ignored in favour of experimental simplicity. Therefore, it is crucial to experimentally investigate the magnitude of stress concentration in composite specimens due to geometric non-linearity at different testing parameters.

In this study, woven composite material was tested to investigate its quasi-static and fatigue behaviour. Past studies have investigated the fatigue life, delamination behaviour, and fracture toughness of woven composites due to tension and compression loads [19–23]. The damage mechanisms were also studied using non-destructive techniques such as thermography, ultrasonics, and X-ray tomography [24–27]. Additionally, open-hole woven composite specimens were tested to examine the notch sensitivity and notch size effects on the fatigue behaviour of the material. The strain concentration around the notch region was shown to be high by S.Dai et.al., by using digital image correlation technique and further concluding that the notch size has a negligible influence on the quasi-static tensile strength [28]. Similarly, other researchers confirmed the negligible effect of open-hole size on the fatigue sensitivity and failure strength of woven composites [29–31]. The effect of geometric nonlinearity in a dog-bone specimen is similar to the notch effect in an open-

hole specimen. Hence, this strain concentration is expected on the dog-bone specimen edges; however, the radius of the edge curvature may have a negligible impact on the strain concentration.

## 2. Materials and Specimen Preparation

The composite laminates were fabricated using AX-5112T woven carbon fibres pre-impregnated with epoxy matrix, supplied by Axiom Materials Inc, Santa Ana, CA, United States. The prepregs were vacuum bagged under $-25$ psi vacuum and cured in an autoclave at a dwell temperature of 121 °C for 240 min, under a uniform pressure of 72 psi, to obtain the composite laminate. The cured rectangular laminates were cut into large dog-bone fatigue specimens using high-pressure abrasive waterjet cutting, as recommended in the literature [32]. The cut cross-section of the specimen was observed via an optical microscope to ensure the absence of delamination due to the cutting process. A balanced and symmetric composite layup, $[(0/90)_1/(+45/-45)_4]_s$ with an average cured thickness of 1.7 mm was investigated in this study. The fatigue specimens were bonded with $[0/90]_{7s}$ CFRP tabs using the Redux 609 epoxy adhesive film. Two adhesive films were used to bond each tab surface to the specimen surface. The specimen and tab surfaces were sanded before bonding to improve the adhesion. After bonding, the specimens were cured in an autoclave at 120 °C for 60 min under 40 psi uniform pressure. The temperature was ramped up and down at 10 °C/min during both the heating and cooling cycles, respectively.

## 3. Methodology

The large dog-bone shaped composite specimens were tested on a hydraulic testing machine with 100 kN load cell capacity, supplied by MTS, Eden Prairie, MN, United States. The specimens were housed in a specially designed fixture to enable the application of large compressive loads without buckling. The fixture shown in Figure 1, consists of six-bolt connections that fasten the specimen to the loading actuators of the MTS machine. The high torque applied to these bolts ensures that the load is transferred through shear, thus avoiding any bearing stresses. The bearing stresses around the specimen bolt holes were kept to a minimum during testing, evidenced by the absence of bearing failure in any of the specimens. The fixture has a cage-like structure around the gauge section of the specimen, which houses fourteen anti-buckling rods that prevent buckling of the specimen during compression loading. The anti-buckling rods are in soft contact with the specimen surfaces to prevent buckling. To avoid surface abrasions, the friction between the rods and the specimen was kept to a minimum by adjusting the pressure exerted by the rods on the specimen.

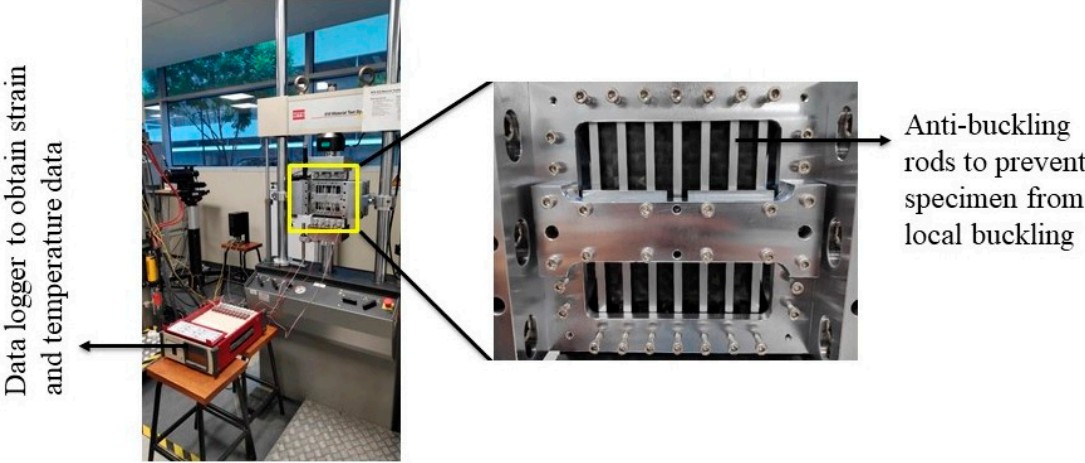

**Figure 1.** Composite specimen housed in the anti-buckling fixture.

Although the dimensions of the fatigue specimen are proprietary, and thus not mentioned in this paper, the outline of the specimen geometry is as shown in Figure 2. Two GFLAB-3-350-70 strain gauges were mounted at the centre and edge of the composite specimen, as shown in Figure 2. Similarly, two thermocouples were mounted on top of each strain gauge to measure the temperature increase during fatigue testing. The strain gauges and thermocouples were connected to a TDS-530 data logger, supplied by Tokyo Measuring Instruments Laboratory Co., Ltd., Tokyo, Japan, to acquire the strain and temperature data at 1 Hz frequency.

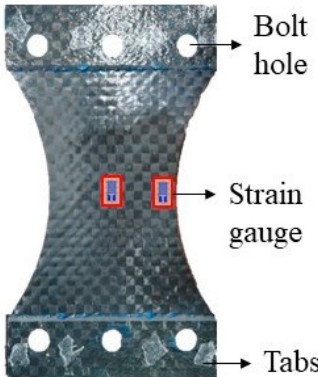

**Figure 2.** Dog-bone shaped composite specimen with centre and edge strain gauges (SG1 and SG2).

The composite specimens were subjected to both quasi-static and fatigue loadings to investigate the respective behaviour at room temperature. Quasi-static tension and compression loads were applied to the specimens at a displacement rate of 0.5 mm/min. Each specimen was subjected to quasi-static loads within the elastic limit (up to 15 kN) without inducing any lasting damage. The objectives of the test were to estimate Young's modulus (E) of the material and to measure the centre and edge strains of the specimens at their respective locations. After the quasi-static test, each specimen was subjected to four fatigue load cases, as shown in Table 1.

**Table 1.** Fatigue load cases.

| Fatigue Load Case | Stress Ratio | Frequency (Hz) | Maximum Load (kN) | Minimum Load (kN) |
|:---:|:---:|:---:|:---:|:---:|
| A | −0.1 | 3 | 20 | −2 |
| B | −0.1 | 5 | 20 | −2 |
| C | −0.5 | 3 | 20 | −10 |
| D | −0.5 | 5 | 20 | −10 |

During the quasi-static tension test, the anti-buckling rods had a very marginal contact with the specimen. However, during the quasi-static compression and T-C fatigue loading, the pressure on the specimen by the anti-buckling rods was increased to prevent buckling. Since two strain gauges were mounted on the specimen to measure the strains, two anti-buckling rods near the strain gauges had to be removed during testing. To prevent local buckling around this region, slotted anti-buckling rods were used to make room for strain gauge installation (see Section 4.1 for more details).

## 4. Results and Discussions

### 4.1. Quasi-Static Results and Critical Buckling

To investigate the possibility of local buckling in the specimens during compressive loading, three different scenarios were considered. In the first case, two buckling rods on the front surface of the specimen, where the strain gauges were mounted, were removed,

as shown in Figure 3. In the second case, two rods from the front and back surfaces, respectively, were removed. Therefore, in the first and second cases, a total of 12 and 10 anti-buckling rods prevented buckling in the specimens, respectively. In the final case, the two rods in the front face were replaced by slotted rods that allowed the mounting of strain gauges in the gap created by the slots. The seven rods on the back face were not removed or replaced in the third case.

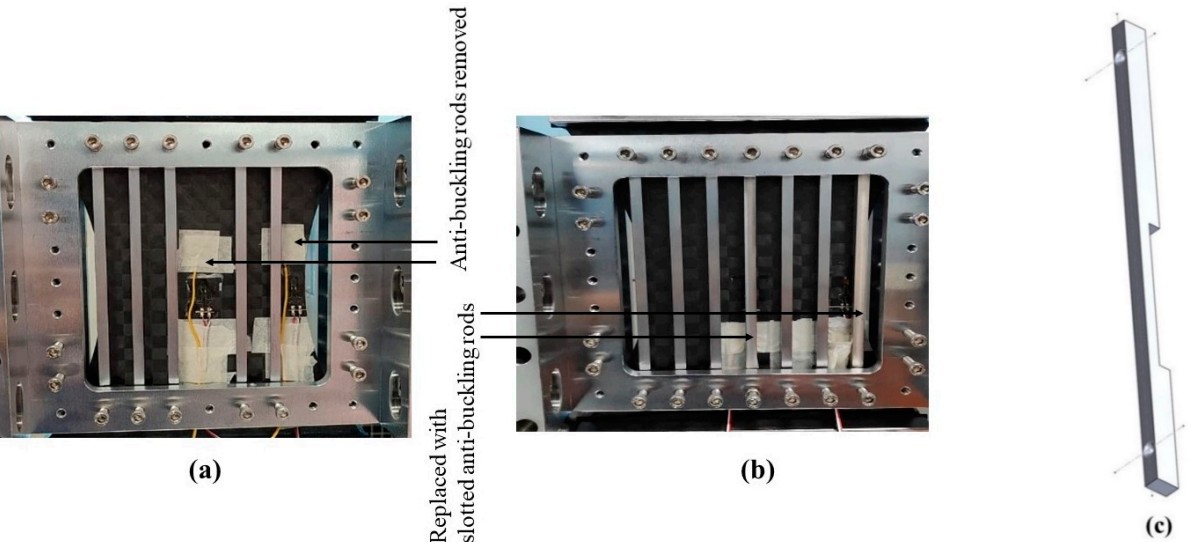

**Figure 3.** (**a**) Two ordinary anti-buckling rods removed from the fixture to mount the strain gauges (center and edge); (**b**) Two slotted anti-buckling rods are used to create a gap to mount the strain gauges; (**c**) Slotted anti-buckling rod.

Two, four and two strain gauges were mounted on the specimens belonging to cases 1, 2 and 3, respectively. The strain readings due to quasi-static and fatigue loadings were compared for all the cases to identify the possibility of buckling and the corresponding critical buckling loads. The positions and nomenclature of the strain gauges mounted on the specimens for all the cases are as shown in Figure 4.

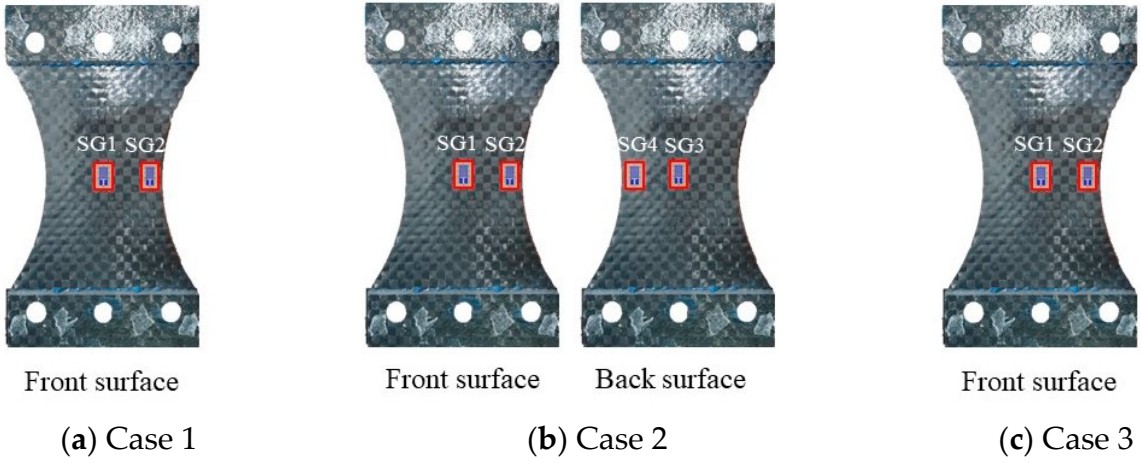

**Figure 4.** Strain gauges mounted on the specimens: (**a**) Case 1—without ABRs on the specimen front surface; (**b**) Case 2—without ABRs on both the front and back surfaces; (**c**) Case 3—with slotted ABRs on the front surface. (ABRs: Anti-Buckling Rods).

The occurrence of local buckling in case 2 is evident from Figure 5, where the strain differences at the centre (SG1-SG3) and edge (SG2-SG4) of the specimen are plotted with respect to the applied quasi-static loads. Since buckling does not occur during tensile loading, the strain differences are due to the inherent differences in the strain gauges.

However, during compressive loading, the strain differences drastically increase after a 10 kN static load due to the occurrence of local buckling. This represents the critical buckling load of the specimen due to quasi-static compression loading.

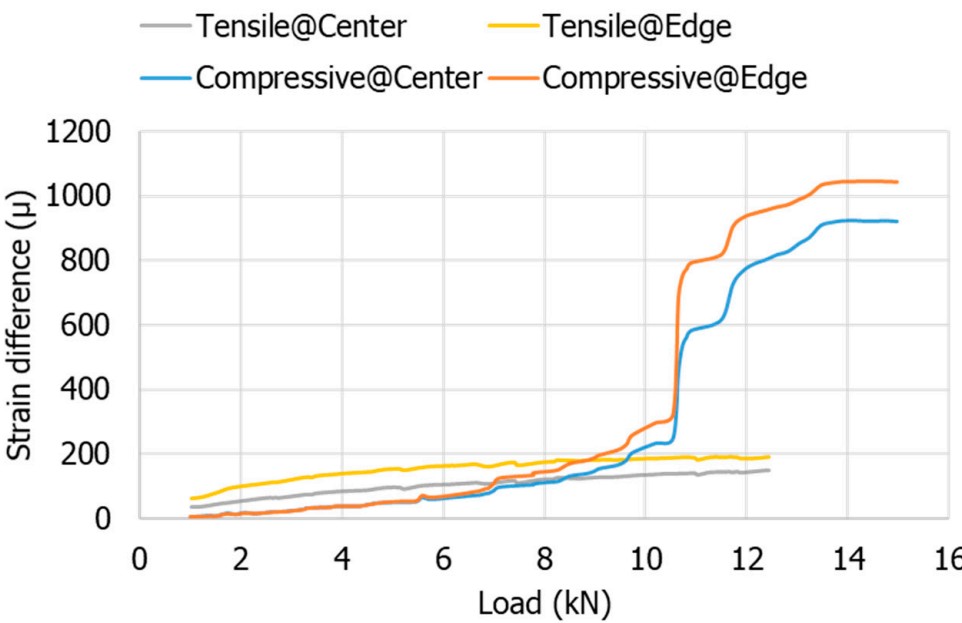

**Figure 5.** Difference between the edge and centre strains of the specimen due to quasi-static compressive loads.

The occurrence of local buckling could be avoided by using slotted anti-buckling rods instead of removing the normal anti-buckling rods, as conducted in cases 1 and 2. Figure 6 compares the centre and edge compressive strains of the specimens in cases 2 and 3, with respect to time. Since a local buckling is experienced in case 2, the compressive strain rate increases on the front face after 50 s (due to buckling-induced compression), while it decreases on the back face (due to buckling-induced tension). However, the front face of case 3 does not experience buckling-induced compression due to the presence of the slotted anti-buckling rods. Hence, the compressive strain rate remains constant during the quasi-static tests.

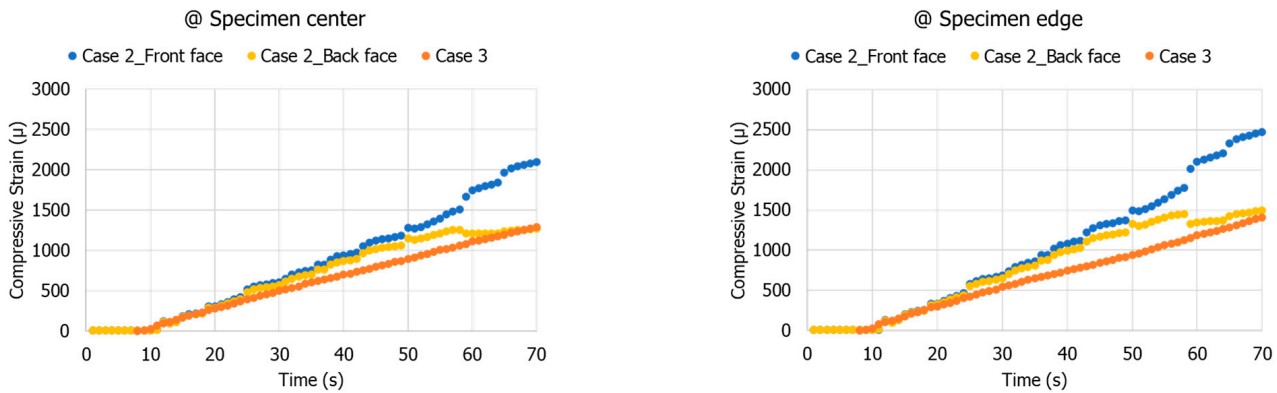

**Figure 6.** Effect of slotted anti-buckling rods on the compressive strains of the specimens due to quasi-static loads.

Since the slotted anti-buckling rods prevented local buckling, the composite specimens were tested by following case 3. The tensile and compressive Young's moduli (E) of the material were found to be $34.26 \pm 2.43$ GPa and $34.20 \pm 1.89$ GPa, respectively, while its tensile strength was experimentally estimated as $322 \pm 7.9$ MPa by following ASTM D3039

standard. The strength of the material was estimated using rectangular samples measuring 110 mm × 20 mm × 1.7 mm. Figure 7 depicts the stress-strain graph of the material under quasi-static tension and compression loads.

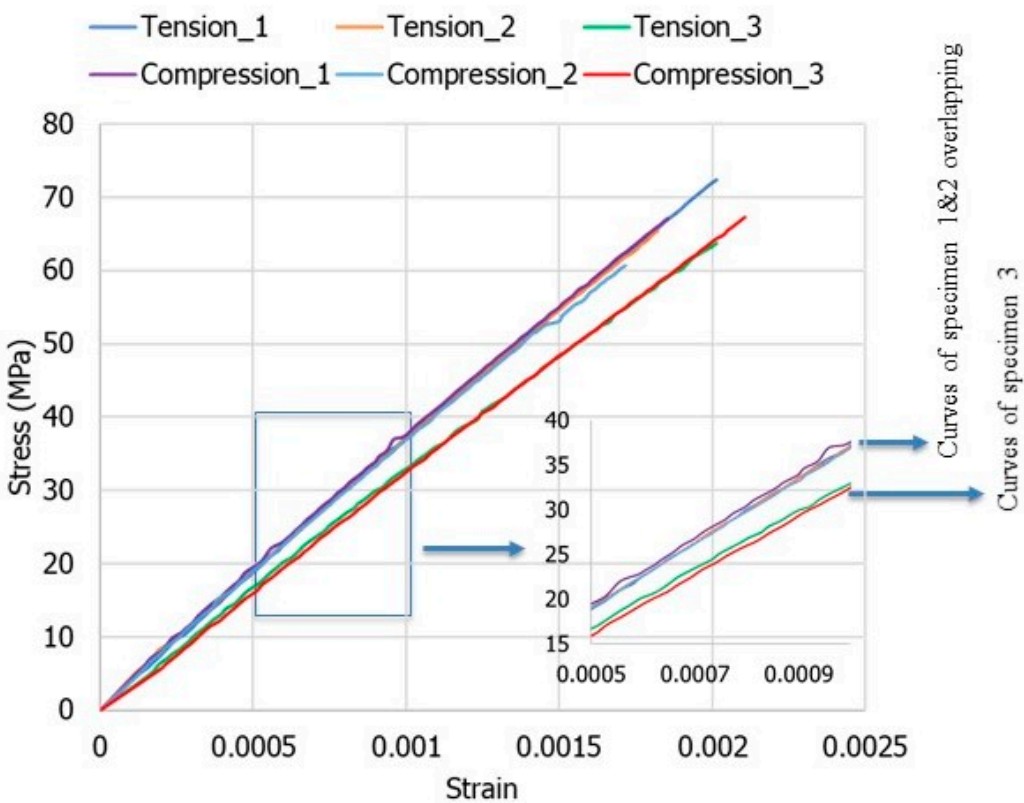

**Figure 7.** Stress-strain variation due to quasi-static tension and compression loads.

The strain concentration on the edges of the composite specimens due to quasi-static loading is as shown in Figure 8, where the strains at the edges of the specimens are plotted against the strains at the centre. From the slope, it is evident that the strain concentration due to tension is significant compared to compression. In fact, the opposite effect was observed in the third specimen, where the centre compressive strain was less compared to the edge compressive strain. On average, the strain concentration due to quasi-static tension is 15% and considered significant. It is insignificant at 3% due to quasi-static compression. The quasi-static behaviour of the material is similar under tension and compression loads, as evidenced by the equal magnitude of Young's moduli, due to both types of loadings. However, in quasi-static tension, the occurrence of strain concentration at the edge of the specimen distinguishes the tensile and compressive behaviour of the material. Assuming a constant material stiffness, the strain concentration at the specimen edge introduces a stress concentration of a similar magnitude, which can be quantified using Hooke's law.

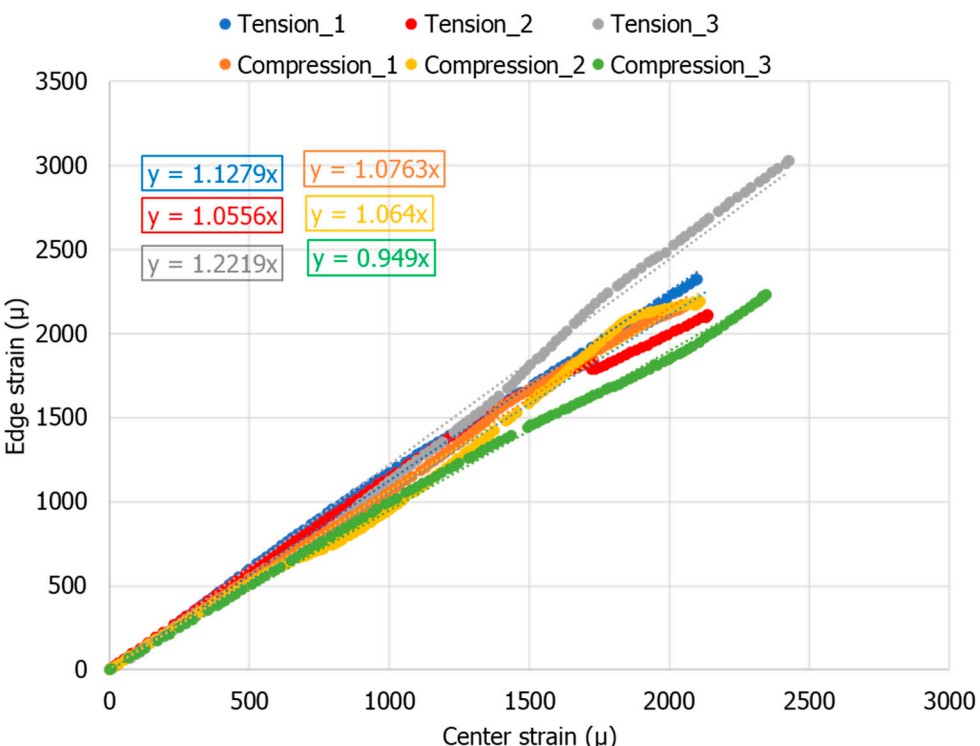

**Figure 8.** Stress concentration due to quasi-static tension and compression loads.

### 4.2. Fatigue Results

The maximum compressive fatigue load applied in this study was chosen as −10 kN based on the quasi-static critical buckling load. Additionally, slotted anti-buckling rods were used to avoid buckling during the fatigue tests. The strains at the specimens' centre and edge, due to the four fatigue load cases, were first acquired using the strain gauges and the data logger. The centre and edge strains were compared to quantify the strain concentration on the edges due to the four different load cases. The corresponding stress concentration at the specimens' edges was estimated using Hooke's law (assuming constant E).

The testing frequency was greater than the data acquisition frequency in this study. Hence, the graphs plotting the edge strains against the centre strains were dependent on the difference between the testing and data acquisition frequencies. Figure 9 depicts the difference in the graphs due to the applied loading frequency. Since the data acquisition frequency was less than the testing frequency, enough data points were not obtained to achieve an ideal linear plot (similar to Figure 8 for the quasi-static load case). The slope of the linear plot would have represented the strain concentration factor due to fatigue loading. However, at a 5 Hz loading frequency, an ellipse is observed while plotting the edge strains against the centre strains. The ellipse represents the periodic nature of loading and data acquisition, while the aspect ratio of the ellipse (ratio of the major axis to the minor axis) depicts the difference between the loading and data acquisition frequencies. As the loading frequency is reduced to 3 Hz and 1 Hz, the aspect ratio of the ellipse increases and tends towards the ideal linear plot. If the increase in the data acquisition rate is higher than the loading frequency, enough data points can be collected to obtain the ideal linear plot. However, in the current study, the data acquisition rate was kept at 1 Hz due to the instrument constraints. Although the aspect ratio increases with the reducing loading frequency, the maximum and minimum strains due to fatigue loading remain almost constant irrespective of the differences between the loading and data acquisition frequencies. Hence, this was used to calculate the strain difference (between the centre and the edge) due to the maximum and minimum loads during the fatigue loading.

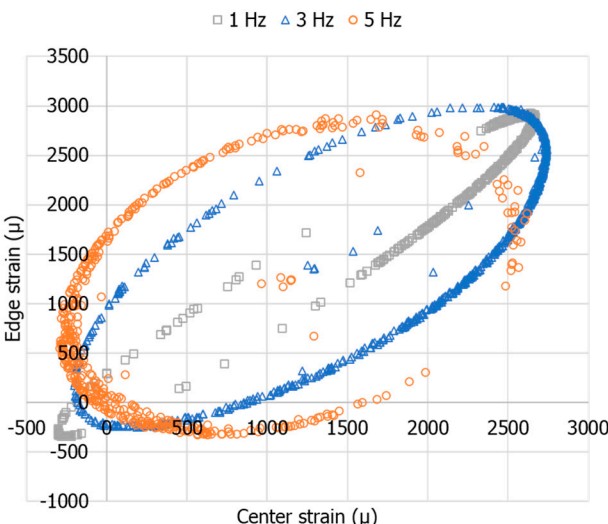

**Figure 9.** Effect of fatigue loading frequency on the edge vs. centre strains.

The effects of loading frequency and stress ratio on the strains of the specimens are depicted in Figure 10, where the edge strains are plotted against the centre strains due to the four different fatigue load cases. The corresponding stresses are also plotted.

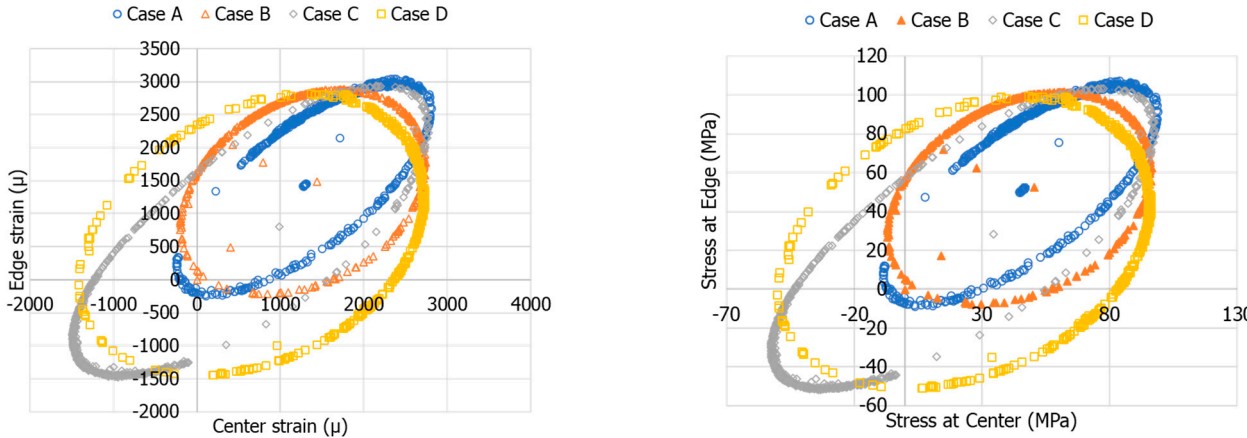

**Figure 10.** Stress and strain plots due to the four fatigue load cases.

From the plots displayed in Figure 10, the maximum and minimum strains and stresses were obtained for the four fatigue load cases. Figure 11 displays the average stresses and strains at the centre and edges of three fatigue specimens, each tested at the four fatigue load cases. Furthermore, the stress concentration percentages at the edges of the specimens were calculated from the data displayed in Figure 11.

From Figure 12, it is evident that the stress concentration at the edges of the specimens due to the tensile part of the fatigue load is almost constant irrespective of the stress ratio and frequency. This is because the same tensile load is applied in all four load cases and the frequency seems to not affect the stress concentration. However, the stress concentration due to the compressive part of the fatigue load is significantly different for the four fatigue load cases. For cases A and B, the applied compressive load is marginal at −2 kN; hence, the probability of measuring the corresponding strains and stresses using the data logger during fatigue loading is significantly lower compared to cases C and D, where the applied load is −10 kN. Hence, the stress concentration percentages for cases C and D are more reliable than for cases A and B. This suggests that the stress concentration is high due to the tensile load during tension-compression fatigue (at a high stress ratio only) than due

to compressive load. A similar observation was also made while comparing the stress concentration at the specimen edges due to quasi-static tension and compression loads.

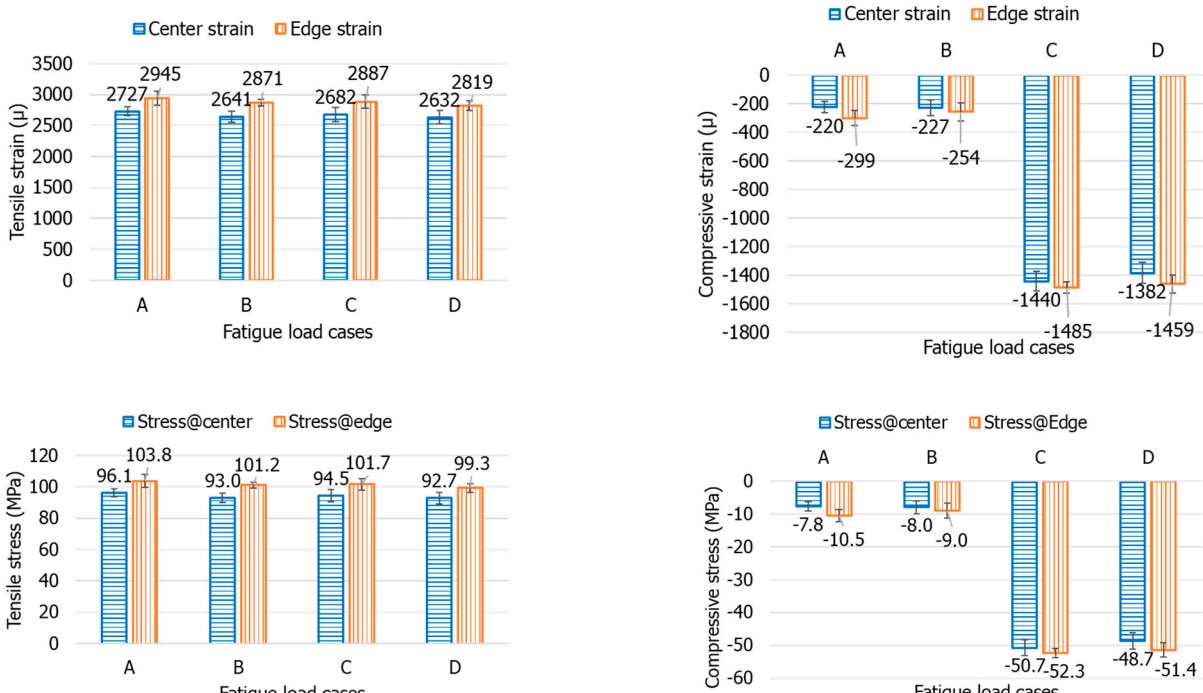

**Figure 11.** Comparison of the centre and edge stress and strain due to the four fatigue load cases (A–D).

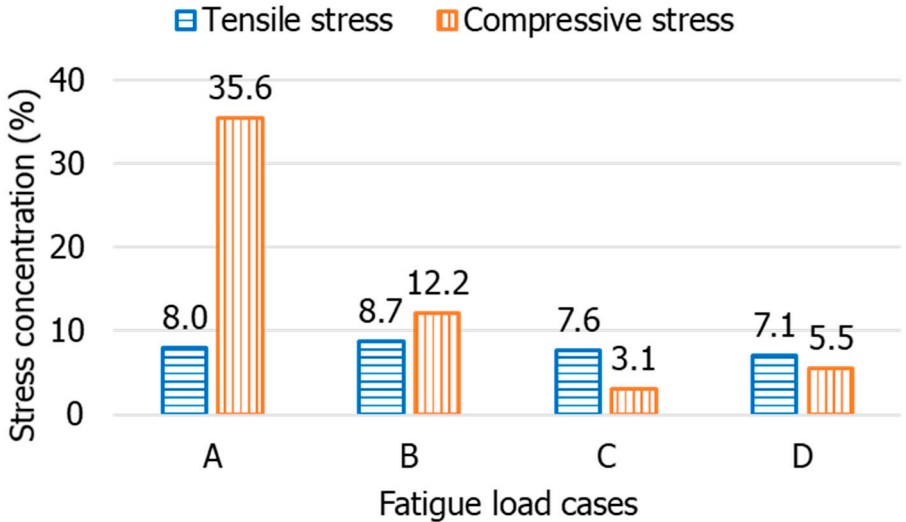

**Figure 12.** The magnitude of the stress concentration at the specimen edges due to the four fatigue load cases.

The consequence of stress concentration at the edge of the specimen is the initiation of delamination between the load-carrying cross ply (0/90) and the angle ply (+45/−45), as shown in Figure 12. The stiffness mismatch between the cross ply and the angle ply is responsible for the delamination initiation, which is then propagated due to the tension-compression fatigue load [33,34]. This stiffness mismatch is aggravated at the edge due to the stress concentration effect. Eventually, the delamination results in the catastrophic damage of the fatigue specimen at a higher load (75% of the quasi-static ultimate load), as shown in Figure 13. Prior to catastrophic failure, the cross ply at the edges of the specimen

experiences lateral cracks due to stress concentration. Following the lateral crack, the angle plies in the centre of the specimen fail in shear.

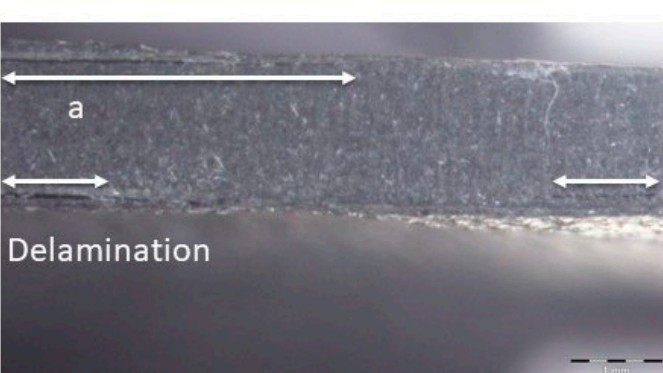 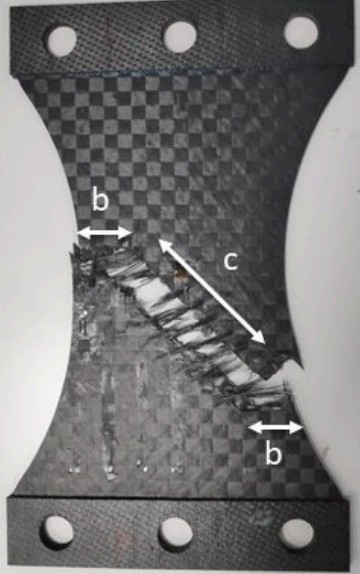

**Figure 13.** Fracture behaviour: (a) Delamination due to stiffness mismatch between the cross ply and angle ply; (b) Initiation of the lateral crack of the cross ply due to the stress concentration; (c) Shear failure of the angle plies follows the lateral failure of the cross ply.

## 5. Conclusions

The tension-compression fatigue behaviour of woven carbon-fibre-reinforced epoxy composite material was studied by considering the size effects, loading frequency, stress ratio, and geometry-induced stress concentrations. Slotted anti-buckling rods were used during the quasi-static and fatigue loadings to allow the mounting of strain gauges at the centre and edges of the test specimens, while simultaneously avoiding local buckling. The following inferences were drawn from the experimental investigation performed in this study.

- In the absence of some of the anti-buckling rods (at the centre and edge of the specimen to mount the strain gauges), the critical buckling load during quasi-static compression was identified as −10 kN. This suggests the occurrence of local buckling in the specimen due to the removal of the rods. Hence, to avoid buckling during fatigue loading, the upper compressive load limit was set at −10 kN and the slotted anti-buckling rods were used during loading.
- The slotted anti-buckling rods effectively prevented buckling from occurring for up to a minimum of a −15 kN compressive load (maximum applied load in this study).
- The strain concentration at the edges of the specimens due to quasi-static tension and compression was 13.5% and 9%, respectively. This suggests that the strain concentration, and subsequently the stress concentration, is significant due to quasi-static tension compared to quasi-static compression.
- At a high stress ratio, the tensile segment (7.35%) of the tension-compression fatigue loading has a significant effect on the stress concentration than the compressive segment (4.3%). Additionally, the loading frequency (0.1%) and stress ratio (1%) have a negligible effect on the stress concentration due to tension.
- Furthermore, the stress concentration percentages due to compression are more reliable at a higher stress ratio than at a lower stress ratio due to insufficient data acquisition rate.

**Author Contributions:** Conceptualization, E.M. and R.A.N.; Data curation, E.M. and R.A.N.; Formal analysis, R.A.N.; Funding acquisition, S.C.J., P.A. and P.M.; Investigation, E.M. and R.A.N.; Methodology, E.M., R.A.N., S.C.J., P.A. and P.M.; Project administration, S.C.J., P.A. and P.M.; Resources, P.A. and P.M.; Supervision, S.C.J.; Writing—original draft, E.M. and R.A.N.; Writing—review & editing, E.M., R.A.N., S.C.J., P.A. and P.M. All authors have read and agreed to the published version of the manuscript.

**Funding:** This research received no external funding.

**Institutional Review Board Statement:** Not applicable.

**Informed Consent Statement:** Not applicable.

**Data Availability Statement:** The data presented in this study are available on request from the corresponding author.

**Acknowledgments:** The authors gratefully acknowledge Nanyang Technological University and Leonardo company for all the support during the study.

**Conflicts of Interest:** The authors report no conflict of interest.

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
