# Peer review of "Tension-Compression Fatigue Induced Stress Concentrations in Woven Composite Laminate"

_jcs, doi:10.3390/jcs5110297_

Round 1

Reviewer 1 Report

Authors have examined an interesting topic in the field of fatigue and laminate composites. Experimental results are filling the gap regarding the buckling behavior of large specimens. Some points cited below can be modified/corrected for the sake of the improvement of the manuscript.

  • Line 154 – Bucking > Buckling
  • The curves in Figure 7 are difficult to distinguish due to color selection. Colors of the curve can be modified and an inset can facilitate the understanding of the figure.
  • Interpretation of Figure 8 can be enriched.
  • The argument given in Line 188-189 should be explained more clearly. It is a bit ambiguous.
  • The root cause of the changing aspect ratio of the ellipsoid with respect to testing frequency should be explained in detail.

Author Response

Response to reviewers comments is attached in the word document below.

Reviewer 2 Report

This research focuses on the investigation of the fatigue damage induced in tensile specimens subjected to tensile and compression load during alternate tests considering two stress ratios. The topic is well-known in the literature and the content of the manuscript does not address any new issue, so it lacks novelty. Moreover, although the research has been carried out properly, from the specimen preparation to the results organization, it lacks the scientific background required for an academic publication.

While reading the manuscript, a few points that can be addressed to improve the manuscript came to my mind, as follows:

1) The literature review is quite limited. Plenty of work on analytical modeling and experimental investigation has been carried out on these topics in the last 20+ years. I suggest the authors to look for additional reference works dealing with fatigue and woven composite and integrate them in the introduction. It might be helpful to look for papers of the research group of Prof. Marino Quaresimin and Prof. Michele Zappalorto of the University of Padua.

2) The issue of cutting cured specimens from plates has been discussed in the literature since the cutting process (waterjet, laser, machining, etc) can create weak spots in the specimen surface, which may act as crack initiation points. In section 2 authors state “The cured rectangular laminates were cut into large dog-bone fatigue specimens using waterjet cutting”. I suggest the authors to critically address the above-mentioned issue by either citing similar works where the same process has been utilized or to underline the countermeasures they employed to assure that the cutting process did not have any effect on the specimens’ mechanical performances.

3) In the results section it would be interesting to compare the stress concentration with the fracture behavior of the material, for instance by taking some SEM pictures of the fracture area and try to correlate these images to the different notch sensitivity. Try to investigate the experimental results you collected from both the analytical and experimental points of view in order to provide a more critical discussion of the findings.

Author Response

(The authors gave the same response as above.)

Reviewer 3 Report

The performed practical work studied stress concentrations due to T-C fatigue in the woven dog-bone composite specimen. The composite structure was fabricated by woven CFRP. Specimen were designed in both anti-buckling and buckling forms. It was shown that the stress concentration at the edges of the specimen during tension is further critical than compression.

I found this work interesting and I believe it suits the journal. Thus, I recommend the manuscript for publication after a minor modification based on the following comments,

-Line 145 should be checked. The reference does not exist.

-Quality of figures 7-10 is insufficient.

Round 2

Reviewer 1 Report

The authors examine an important topic and after taking reviewers' suggestions they have improved the manuscript.

Reviewer 2 Report

Although the authors improved the manuscript following my suggestions, I still feel that the manuscript, at the present state, lacks the proper academic relevance required for a sceintific paper.

The auhtors came a great lenght in designing and carrying out the experiments presented in the paper but, experimental results alone, on a topic such that of fatigue of woven composites, are not enough for the research to become academically relevant. In a nutshell, the manuscript, at the present state, lacks novelty in comparison to previously published contributions.

If the authors aim to improve the manuscript, I suggest to include either a theoretical or FEM investigation of the state of stress developing in the material and, if possible, propose a fatigue model correlating the stress state with the fatigue life. Obviously, the model has to be compared with previous literature models, in order to demostrate the improvements. These are marely some simple suggestions that I felt to convey to the authors to provide some hint about how they can improve the manuscript.

Reviewer 3 Report

The paper in the current form can be recommended for publication.